# Genetics of Obesity in Humans: A Clinical Review

**DOI:** 10.3390/ijms231911005

**Published:** 2022-09-20

**Authors:** Ranim Mahmoud, Virginia Kimonis, Merlin G. Butler

**Affiliations:** 1Department of Pediatrics, University of California, Irvine, CA 92697, USA; 2Department of Pediatrics, Faculty of Medicine, Mansoura University, Mansoura 35516, Egypt; 3Departments of Neurology and Pathology, University of California, Irvine, CA 92697, USA; 4Children’s Hospital of Orange County, Orange, CA 92868, USA; 5Departments of Psychiatry & Behavioral Sciences and Pediatrics, University of Kansas Medical Center, Kansas City, KS 66160, USA

**Keywords:** obesity, genetics, monogenic, polygenic, Prader-Willi, syndrome

## Abstract

Obesity is a complex multifactorial disorder with genetic and environmental factors. There is an increase in the worldwide prevalence of obesity in both developed and developing countries. The development of genome-wide association studies (GWAS) and next-generation sequencing (NGS) has increased the discovery of genetic associations and awareness of monogenic and polygenic causes of obesity. The genetics of obesity could be classified into syndromic and non-syndromic obesity. Prader–Willi, fragile X, Bardet–Biedl, Cohen, and Albright Hereditary Osteodystrophy (AHO) syndromes are examples of syndromic obesity, which are associated with developmental delay and early onset obesity. Non-syndromic obesity could be monogenic, polygenic, or chromosomal in origin. Monogenic obesity is caused by variants of single genes while polygenic obesity includes several genes with the involvement of members of gene families. New advances in genetic testing have led to the identification of obesity-related genes. Leptin (*LEP*), the leptin receptor (*LEPR*), proopiomelanocortin (*POMC*), prohormone convertase 1 (*PCSK1*), the melanocortin 4 receptor (*MC4R*), single-minded homolog 1 (*SIM1*), brain-derived neurotrophic factor (*BDNF*), and the neurotrophic tyrosine kinase receptor type 2 gene (*NTRK2*) have been reported as causative genes for obesity. NGS is now in use and emerging as a useful tool to search for candidate genes for obesity in clinical settings.

## 1. Introduction

Obesity is a major health problem worldwide. It is more common in established countries but is on the increase in developing countries. The worldwide prevalence of obesity [body mass index (BMI) ≥ 30 kg m^2^] has doubled between 1980 and 2008, with a prevalence of 13% in the adult population reported in 2014. In 2013, there were 42 million obese children below the age of five years [1]. BMI, according to the World Health Organization (WHO), is classified in adults as overweight at 25–29.9 kg/m^2^, obese at 30–39 kg/m^2^, and morbidly obese at 40 kg/m^2^ and above [2].

Obesity is a complex multifactorial disorder with genetic and environmental factors. The increased prevalence of obesity is impacted by the environment as high caloric food sources with a sedentary lifestyle has decreased energy expenditure. Twin and family studies have documented the role of genetic factors in obesity, with the risk of childhood obesity increasing with a positive family history of obesity. There is a high concordance rate for obesity in monozygotic twins vs dizygotic twins and an estimated heritability for obesity at between 40% and 75% in twin studies [3].

The recognition of obesity and inheritance with associated genes has been impeded by a limited knowledge and understanding of genetics at the human genome level and with the biological pathways involved in obesity. However, the development of genome-wide association studies (GWAS) and next-generation sequencing (NGS) has increased the discovery of genetic associations and awareness of monogenic and polygenic causes of obesity. About 127 informative sites in the human genome have been reported to show linkage with obesity by GWAS [4] and over 500 obesity-related genes recognized in humans [5]. There are approximately 30 neuro-endocrine peptides in humans that are known to inhibit eating behavior, but only ghrelin increases eating with an important role in appetite regulation and energy balance [6]. This balance is in response to changes in peripheral circulating signals from adipose tissue, stomach, and endocrine organs. Regions of the brain and neurons help with energy balance and homeostasis by sensing and processing various metabolic signals with major activity observed in the hypothalamus. Many monogenic neuroendocrine disorders involving the leptin pathway are recognized and associated with early onset obesity in childhood. The genetics of obesity could be classified into syndromic and non-syndromic obesity with or without congenital defects and developmental delay. For example, Prader–Willi, fragile X, Bardet–Biedl, Cohen, and Albright Hereditary Osteodystrophy (AHO) syndromes are associated with developmental delay and early onset obesity [7]. Non-syndromic obesity could be monogenic, polygenic, or chromosomal in origin. Monogenic obesity is caused by variants of single genes while polygenic obesity includes several genes with the involvement of members of gene families with or without syndromic findings but accompanied with obesity and recognized phenotypes.

New advances in genetic evaluation and analysis have led to the identification of obesity-related genes. For example, eight genes have been reported as causes for obesity, including leptin (*LEP*), the leptin receptor (*LEPR*), proopiomelanocortin (*POMC*), prohormone convertase 1 (*PCSK1*), the melanocortin 4 receptor (*MC4R*), single-minded homolog 1 (*SIM1*), brain-derived neurotrophic factor (*BDNF*), and the neurotrophic tyrosine kinase receptor type 2 gene (*NTRK2*) [8,9], from over 500 obesity-related genes [5]. One important method is termed GWAS, which incorporates hundreds, or thousands, of polymorphic DNA markers and single nucleotide polymorphisms (SNPs) located throughout the human genome with the ability to search for markers for new genes with no previous evidence of disease involvement. In the past decade, close to 1000 published GWAS results have been reported and 165 traits found in humans with a number of SNPs from obese and nonobese individuals marked gene loci and potential candidate obesity genes [10]. Hence, genetic factors can be divided into the following three categories: Mendelian (monogenic) syndromic obesity, Mendelian non-syndromic obesity, and polygenic obesity. Meanwhile, NGS is now in use and emerging as a useful tool to search for candidate genes for obesity in clinical settings [5,11]. The results of these recent investigations need to be replicated to warrant further consideration.

## 2. Obesity-Related Genes and Defects

### 2.1. Leptin

Leptin is a protein secreted by white adipose tissue and encoded by a gene on chromosome 7 in humans. Leptin crosses the blood–brain barrier to bind to the presynaptic GABAergic neurons of the hypothalamus and decreases appetite and increases energy expenditure [12]. In the arcuate nucleus of the hypothalamus, leptin binds to its receptor and inhibits the neuropeptide Y (NPY)/agouti-related protein (AgRP) pathway [13]. The role of leptin and the leptin receptor gene in human obesity is now emerging but not well understood [14]. Farooqi (2005) reported that inherited human leptin deficiency in patients caused severe early-onset obesity (e.g., 8 years and 86 kg, or 2 years and 29 kg) due to a frame-shift mutation in the homozygous obesity leptin gene (deletion of G133) and a truncated protein [7]. Other studies reported a high level of leptin in obese patients, but it was associated with a decrease in the level of soluble leptin receptors, which contributed directly to leptin function. Another study on 110 patients including 55 obese and 55 healthy controls showed significantly higher levels of leptin in the obese group than in controls [4]. This phenomenon is known as leptin resistance. These receptors are not only found in the CNS but are also present in peripheral organs, such as the liver, skeletal muscles, pancreatic beta cells, and even adipose cells, thereby playing an important role in energy regulation.

### 2.2. Proopiomelanocortin (POMC) Deficiency

*POMC* is an appetite inhibitory gene found on chromosome 2 in humans. It influences the leptin–melanocortin system as a deficiency of the POMC protein causes an absence of ACTH and alpha-MSH, which are cleaved from the POMC protein [15]. Hence, a deficiency of POMC leads to hyperphagia, lower resting metabolic rate, and resultant severe obesity with red hair and pale skin [16]. Errors in the cleavage of master proteins such as *POMC* require pro-hormone convertase, which cleaves this large protein into smaller functional peptides and as noted, interacts with appetite control, pigment, and obesity [17].

### 2.3. Melanocortin-4 Receptor

The melanocortin-4 receptor *(MC4R*) gene is now considered the most common associated gene for childhood obesity and found in about 4% of affected cases prior to advanced genetic testing and next generation sequencing (NGS) [9]. It was first discovered to be related to body weight in 1998 and now multiple studies have investigated its mechanism and the function of different mutations [18]. The *MC4R* gene codes for the MC4R protein, which plays an important role in energy homeostasis and food intake behavior [19]. The central melanocortin pathway regulates energy balance and homeostasis by activating or inhibiting leptin and its receptor is mediated by two subsets of neurons as well as MC3R and MC4R in the arcuate nucleus of the hypothalamus.

### 2.4. FTO (Fat Mass and Obesity Associated Gene)

*FTO* was the first obesity-susceptibility gene discovered through GWAS in European patients with type 2 diabetes [20]. Multiple single nucleotide polymorphisms (SNPs) in the first intron of the gene have shown a significant association with type 2 diabetes. However, after controlling for BMI, there was no association with type 2 diabetes, thereby suggesting that the FTO and type 2 diabetes association was mediated through FTO’s effect on BMI. Another study was conducted in Sardinian patients and confirmed the same results. The rs9939609 and rs9930506 SNPs were identified in *FTO* with significant association with BMI [21,22]. Other GWAS studies in European populations have reported several other SNPs located in the same chromosomal location. In addition, significant association between *FTO* SNPs (rs9939609, rs17817449, rs12149832) and BMI was reported in three large studies conducted in Asian populations [23,24,25].

Kalantari et al. (2018) reported that the role of *FTO* gene polymorphisms, a haplotype not a SNP, are close to each other so that they can affect other gene expression through a sequence of the first intron region [26]. The association between *FTO* SNPs with food intake and physical activity was investigated in many studies, which revealed the associations between *FTO* SNPs and increased intake of dietary fat, protein, energy, increased appetite, but decreased satiety. However, *FTO* SNPs were not associated with the level of physical activity. This finding highlighted the importance of physical activity in the modulation of body weight even in those with genetic susceptibility to obesity [27].

Additional studies by Castro et al. on four obesity-related genes (*PPARG*-rs1801282; *PPARGC1A*-rs8192678; *FTO*-rs9939609; *MC4R*-rs17782313) showed that three of the four genes (*PPARG, FTO, MC4R*) had a combined effect on overweight and obesity at an odds ratio of 1.65 (*p* = 0.008) in a large case-control study in the Brazilian population [28]. The same *MC4R* variant (rs17782313) and an *FTO* variant (rs9930506) were significantly associated with obesity in children, reported in multiple separate studies involving thousands of individual subjects, particularly Caucasians and Asians [29]. Further studies in children and adolescents with the same genes (*FTO* and *MC4R*) and variants were reported by Resende et al. in a systematic review of the literature with an association with overweight and obesity [30].

Dastgheib et al. performed a metanalysis involving 13 studies with 9565 cases and 11,956 controls on *MC4R* rs17782313 and 18 studies with 4789 cases and 15,918 controls on *FTO* rs9939609. They found that odds ratios showed significant results indicating that these variants were associated with a higher risk of obesity [29].

Many forms of obesity are thought to be polygenic with variants involved in the same or different genes that act synergistically per individual affecting body weight, composition, and size quantitatively. Polycystic ovary syndrome (PCOS) is a common polygenic metabolic disorder affecting 5–8% of women in the childbearing period. PCOS is defined according to the Rotterdam consensus based on diagnostic criteria to include at least two of the following features: (1) clinical or biochemical hyperandrogenism; (2) oligo-anovulation; and (3) polycystic ovaries (PCO) and excluding similar endocrinopathies. Most women with PCOS are overweight or obese. Many studies investigated the role of genetic contribution for obesity in patients with PCOS [31,32]. Ewens et al. reported five SNPs in *FTO* and two in *MC4R* with significant association with BMI in the PCOS families [33]. Another study by Tu et al. reported association between *LEPR* Lys109Arg (rs1137100) and PCOS susceptibility in 326 Han Chinese patients with PCOS [34].

### 2.5. Chromosomal Defects and Obesity

Syndromic childhood obesity is a rare form of obesity that is part of multiple clinical manifestations. Advanced genetic testing has helped in the detection of structural defects of the chromosome and at the DNA level and has led to the diagnosis of rare and common forms of obesity. The determination of genetic causes of obesity could be helpful for genetic counselling and the selection of appropriate treatment. In addition, Dasouki et al. and Cheon et al. each summarized chromosomal abnormalities with syndromic obesity [35,36]. Kaur et al. reported 79 obesity syndromes described in the literature, with obesity considered to be a cardinal feature in 55 of them, while the prevalence of obesity in the other 24 syndromes was higher than that in the general population. Forty-nine syndromes have been mapped to specific chromosome regions or locations including a causative gene [1]. Some examples of syndromic obesity due to chromosomal defects will be discussed in this review such as Prader–Willi syndrome (PWS), Down syndrome, Bardet–Biedl syndrome, fragile X syndrome, Alstrom syndrome, and Cornelia de Lange syndrome. Table 1 highlights other common causes of obesity syndromes and their clinical and genetic findings.

## 3. Obesity-Related Syndromes

### 3.1. Prader–Willi Syndrome

Prader–Willi syndrome (PWS) is a complex genetic disorder affecting multiple body systems. It occurs in 1 in 10,000 to 1 in 29,000 people, affecting both males and females equally and in all races [6,46]. PWS is characterized by hypotonia, decreased muscle tone, and extreme floppiness as an infant, which leads to feeding difficulty and poor weight gain in the newborn or in infancy. Then, completely on the other end of the spectrum, it progresses after infancy to hyperphagia or excessive food drive, which can lead to obesity in childhood and beyond [47]. PWS is a chromosomal disorder with the region associated with PWS located on the chromosome 15q11.2-q13. Typically, people have two different copies of chromosome 15, one inherited from their mother and one from their father. The paternal copy is important for typical development; if a person has not inherited a copy of this region from their father, such as a paternal 15q11-q13 deletion, PWS occurs. Most genes in the 15q11.2-q13 region include imprinted genes and snoRNAs, which are involved in RNA and protein processing of neuroregulators and hormones. When altered, neuronal development and endocrine function are impacted [48].

There are three different genetic mechanisms by which PWS can occur. The most common genetic etiology of PWS is due to the loss of paternal gene expression in the 15q11.2-q13 region, which accounts for about 70% of all PWS cases, caused by a de novo paternally derived chromosome 15q11.2-q13 deletion [48]. The less common form of PWS, occurring in about 30% of all PWS cases, is caused when an individual inherits both copies of chromosome 15 from the mother, known as maternal uniparental disomy (UPD) [6]. A rare form, occurring in about 3% of PWS cases, is a mutation or defect of the imprinting control center in chromosome 15. Therefore, PWS is due to genomic imprinting errors and disturbances of an epigenetic phenomenon resulting in parent-of-origin gene expression, involving methylation and histone modifications and causing monoallelic expression of specific genes [49].

As PWS is characterized by severe hypotonia in the newborn period causing severe floppiness and difficulty in feeding, it can eventually lead to the placement of a feeding gastric tube (G-tube) directly into the stomach or nasogastric tube for feeding assistance in early infancy. A study in France of 19 infants, who were diagnosed with PWS before two months of age, concluded that hospitalization time and duration of tube feeding were reduced due to very early diagnosis. They also found that multidisciplinary care provided (which included growth hormone treatment given between ages 6 months to just under 2 years old) resulted in only 1 infant becoming obese at age 2.5 years [50]. A cross-sectional study of 42 children with PWS and 9 controls, aged 7 months–5 years, investigated differences in appetite hormones that may explain the development of abnormal eating behavior. They found no significant relationship between eating behavior in PWS and the level of any hormone or insulin resistance, independent of age [51]. Oldzej et al. reported that PWS patients with deletion were significantly heavier than those with UPD [52]. Further, Mahmoud et al. in 2021 concluded from a large cohort of PWS patients that higher BMI scores were present in patients with the deletion subtype compared to UPD [53].

PWS has been classically described as having two clinical stages: poor feeding, with failure to thrive (FTT) in infancy (Stage 1), followed by hyperphagia leading to obesity in later childhood (Stage 2). The identification of these phases has assisted in the diagnosis of individuals affected with PWS. Additionally, a study identified a total of seven different nutritional phases, with five main phases and sub-phases in phases 1 and 2 and concluded the progression of nutritional phases in PWS is more complex than previously recognized [54]. An awareness of the various nutritional phases for parents of newly diagnosed infants with PWS may prevent or possibly slow the early onset of obesity. Those affected with PWS are characterized in later infancy or early childhood with hyperphagia or excessive eating with hyperphagia as a difficult symptom to cope with because of the constant desire to eat, even though the individual may have just eaten. (See Figure 1 of an individual with Prader–Willi syndrome as an example of syndromic obesity).

The source of hyperphagia is believed to be located deep in the brain structure in the hypothalamus, a small gland that has multiple roles. It is both an endocrine gland and a key center for a wide variety of behaviors related to survival. The hypothalamus signals to its close neighbor, the pituitary gland, which acts as a master gland with secretions controlling many other glands to release hormones necessary for growth, metabolism, learning, and memory. The hypothalamus also contains key centers for controlling aggression, body temperature, sexual activity, and food and water intake as well as hunger [55]. For people with PWS, the hypothalamus does not regulate emotions and appetite normally as the brain does not receive/process signals of feeling “full or satisfied” and drives the individual to consume more food or eat as much as possible [56]. The brain of an individual with PWS sends signals that the body is starving, lowers the metabolic rate to conserve energy, and drives the individual to find food and eat as much as possible. This excessive food drive, plus the slowed metabolic rate, leads to rapid weight gain and morbid obesity [57]. Obesity often changes the body structure, causing a shorter torso and larger mid-section appearance. Obesity is a major cause of morbidity due to respiratory disease and non-insulin dependent (type 2) diabetes mellitus with comorbidities [57,58].

### 3.2. Alstrom Syndrome

Alstrom syndrome is a rare obesity-related single gene disorder inherited in an autosomal recessive pattern. The estimated range is from 1 in 500,000 to 1 in 1,000,000 and is due to mutations in the *ALMS1* gene located on chromosome 2p13. The ALMS1 protein has an important role in ciliary function, energy metabolism, and cell cycle control. Li et al. (2007) suggested that the absence of the ALMS1 protein leads to abnormal ciliary formation with Alstrom syndrome classified as one of the ciliopathies due to abnormal ciliary function [59].

More than one hundred different mutations have been reported in the literature in the *ALMS1* gene. The symptoms usually start in infancy and progress during childhood with expanded variability in presentation, which makes the diagnosis challenging. The first clinical manifestations are visual problems, nystagmus, and early blindness due to cone-rod dystrophy. Many endocrine abnormalities are reported to occur in Alstrom syndrome including hypothyroidism, hypogonadotropic hypogonadism in males, hyperandrogenism in females, childhood truncal obesity, hypertriglyceridemia, and insulin resistance with type 2 diabetes mellitus. More than 70% of patients with Alstrom syndrome have congestive heart failure due to cardiomyopathy along with short stature, neurodevelopmental delay, scoliosis and kyphosis, and progressive pulmonary, hepatic, and renal dysfunction with associated complications [60].

### 3.3. Fragile X Syndrome (FXS)

FXS is the most common cause of intellectual disability in males. It affects about 1 in 4000 males in the general population and occurs due to the triplet repeat expansion of CGG repeats greater than 200 in size in the 5′ untranslated region of the *FMR1* gene at chromosome Xq27.3 [61]. The carrier state or the premutation form of this gene occurs when the number of CGG repeats is between 50 and 200. Premutation occurs in females and could expand to a full mutation in the subsequent generation. This mutation leads to the loss of fragile X mental retardation protein (FMRP), a protein that plays an important role in protein translation for neuronal synaptic connections [62].

The common clinical features include intellectual disability, large ears, a narrow head, long face, and prognathism. Joint laxity, mitral valve prolapse, and macroorchidism are also common. Behavioral problems in FXS include anxiety, autistic behavior, self-injury, and compulsive disorders. About 10% of individuals with FXS will have severe obesity, hyperphagia, hypogonadism, or delayed puberty as observed in PWS. This type of FXS patient is termed the Prader–Willi phenotype (PWP) [63]. A large survey of families with FXS reported that the prevalence of obesity in adults with FXS was similar to the general population [64]. Another study conducted by the Fragile X Clinical and Research Consortium reported that patients with FXS had higher weights than in the general population [65]. Choo et al. conducted a longitudinal study on 1223 patients with FXS in different age groups and found an increasing BMI with age and higher BMI Z-scores in adulthood, further supporting obesity as a feature [66].

### 3.4. Down Syndrome

Down syndrome (DS) is one of the most common chromosomal disorders in humans [67]. It occurs in 1:600–700 newborns. The most common cause of DS is the presence of an extra copy of chromosome 21. The other causes are Robertsonian translocations and mosaicism involving chromosome 21. In Robertsonian translocations, the long arm of chromosome 21 is translocated and attached to another acrocentric chromosome. In mosaicism, the meiotic non-division occurs after fertilization and at some point during cell division, a chromosome 21 is lost so that the patient has mosaic DS or now has two cell lineages (one with the normal number of chromosomes, and other one with an extra number 21) [68].

Many reviews have examined obesity in children with developmental disabilities specifically targeting children with physical disabilities, coordination disorder, and intellectual disability [69,70,71]. Many mechanisms have been proposed for the development of obesity in DS including increased serum leptin levels associated with increased appetite as the leptin hormone affects the hunger and satiety centers in the brain, decreases energy expenditure, and decreases physical activity [72,73].

The high risk of obesity in DS could be linked to many factors such as genetic predisposition, hypothyroidism, decreased physical activity, high serum cholesterol and triglycerides, and an abnormal diet. In addition, hypotonia, increased susceptibility to systemic inflammation, decreased metabolic rate, depression, and absence of social and financial support could play a role. Decreased cognitive function could be one of the precipitating factors for obesity as it could affect food choice and level of physical activity. Nordstrom et al. in 2020 compared DS patients with mild and moderate intellectual disability along with their nutritional status and found no significant correlation [74]. Fructuoso et al. in 2018 reported an increase in the level of obesity-associated inflammatory biomarkers galectin-3 and HSP72 in a mouse model of DS [75]. This suggested that increased levels in the adipose tissue leading to low-grade inflammation are important risk factors for the development of obesity in DS.

### 3.5. Bardet–Biedl Syndrome

Bardet–Biedl syndrome is a rare form of syndromic obesity that is inherited in an autosomal recessive pattern. The main clinical manifestations are central obesity, retinal cone-rod dystrophy, postaxial polydactyly, learning difficulties, hearing loss, hypogonadism, and genitourinary abnormalities with renal problems such as polycystic kidney disease.

BBS is associated with high genetic heterogeneity, variable expressivity, and pleiotropy. Twenty-four loci are involved with different types of mutations or variants, which could explain different clinical presentations and findings [67]. The different types of mutations include missense, nonsense, deletions, and insertions/duplications of genes causing Bardet–Biedl syndrome. Bardet–Biedl syndrome is a multisubunit complex with involvement of eight proteins coded by *BBS1*, *BBS2*, *BBS4*, *BBS5*, *BBS7*, *TTC8*, *BBS9*, and *BBIP genes.* Most of the Bardet–Biedl syndrome cases in Europe and North America present with mutations in either *BBS1* or *BBS10* genes. Obesity is a common feature as it affects 89% of BBS cases with an early age onset of 2 to 3 years. Obesity occurs in BBS due to gene mutations that lead to a decrease in the number of cilia and altered neuroendocrine signaling from ciliated neurons to fat storage tissues. These disturbances lead to the dysregulation of appetite with changes in leptin resistance and impaired leptin receptor signaling [76].

### 3.6. Albright Hereditary Osteodystrophy

Albright hereditary osteodystrophy (AHO) is an autosomal dominant genetic disorder due to mutations in the *GNAS1* gene. The clinical manifestations include short stature, brachydactyly, developmental delay, pseudo-hypoparathyroidism, a round face, and early onset obesity [77]. *GNAS* is a complex imprinted locus on chromosome 20q13.11. and many transcripts are produced using alternative promoters and splice sites. Alteration in these transcripts can lead to many clinical disorders or presentations. The GNAS1 gene coding Gα_s_ (stimulatory G-protein alpha subunit) mediates signaling by hormones and ligands that bind to G protein–coupled receptors (GPCRs) for generating cyclic AMP. When mutations occur on the maternally inherited alleles expressed in the thyroid or pituitary glands and the renal proximal tubule, a resistance develops to parathyroid hormone (PTH) and other hormones that signal through the Gα_s_-coupled receptors generating disease (pseudohypoparathyroidism type 1A). When mutations occur on the paternally inherited alleles, the patients develop Albright hereditary osteodystrophy without hormone resistance. The role of genomic imprinting is involved in the development of this genetic disorder [78,79].

The etiology of obesity in AHO is not well known but different theories exist including mutations in *MC4R*, which is transduced by Gsα, and mediated anorexigenic signals from hormones and other neurotransmitters. The loss of such anorexigenic signals through MC4R could produce hyperphagia, but this hypothesis has not been widely studied in AHO individuals with obesity [80,81].

### 3.7. WAGR Syndrome

WAGR syndrome occurs due to deletion at chromosome 11p13 (location of the *WT1* and *PAX6* genes). This syndrome is characterized by predisposition to Wilms tumor aniridia, ambiguous genitalia, and mental retardation (WAGR). Many behavioral and psychiatric disorders have been reported in this syndrome including autism spectrum disorders, attention-deficit disorder, obsessive-compulsive disorder, other anxiety disorders, and depression. WAGR syndrome has been associated with a deletion in the brain-derived neurotrophic factor (*BDNF*) gene in the chromosome 11p13 region, which leads to the obesity phenotype. Although persons with WAGR syndrome typically have low-normal birth weight, marked obesity subsequently develops in a substantial subgroup of patients [82,83]. Many case reports have been described with severe hyperphagia, obesity, and cytogenetic deletions of chromosome 11p *BDNF* gene locus [84,85,86]. Han et al. (2008) conducted a study on 33 patients with WAGR and reported that patients with *BDNF* haploinsufficiency had significantly higher BMIs during childhood, with a 100% prevalence of childhood-onset obesity [87].

### 3.8. Cohen Syndrome

Cohen syndrome is caused by a mutation of the vacuolar protein sorting 13 homolog B (*VPS13B)* gene on chromosome 8q22.2. VPS13B is a transmembrane protein that plays an important role in the development and function of the eye, hematological system, and central nervous system via vesicle-mediated transport and the sorting of proteins within the cells [88]. Cohen syndrome has variable clinical manifestations including progressive retinochoroidal dystrophy and myopia, acquired microcephaly, developmental delay, hypotonia, joint laxity, characteristic facial features with prominent central incisors, truncal obesity, cheerful disposition, and neutropenia. Patients with Cohen syndrome usually suffer from failure to gain weight in infancy and early childhood, but later become significantly overweight in their teenage years with mainly truncal fat accumulation. This change usually occurs very rapidly, with a weight gain of 10–15 kg observed over a short period of time from four to six months [89]. Functional studies have shown that the increased fat accumulation in patients with Cohen syndrome is due to an increased propensity of pre-adipocytes lacking the VPS13B protein to differentiate into fat-storing cells [90].

### 3.9. Smith–Magenis Syndrome

Smith–Magenis syndrome is a genetic condition due to an interstitial deletion of chromosome 17p11.2, which is inherited in an autosomal dominant pattern. Patients with Smith–Magenis syndrome are characterized by mental retardation, developmental delay, renal anomalies, sleep disturbances, dysmorphic features, and behavioral problems including maladaptive/self-injurious, aggressive, and food seeking behaviors like patients with PWS. More than 90% of patients with Smith–Magenis syndrome are overweight or obese after 10 years of age [91].

### 3.10. Kallmann Syndrome

Kallmann syndrome is a rare genetic condition of gonadotropin-releasing hormone deficiency and anosmia. Some patients have some additional anomalies including abnormal eye movements, ptosis, hearing loss, unilateral renal agenesis, cleft lip or palate, and obesity. It occurs due to mutations in *KAL1*, *FGFR1*, *FGF8*, *PROKR2*, and *PROK2* genes and most of the cases are inherited in an X-linked recessive pattern and autosomal recessive or dominant pattern with incomplete penetrance [92].

## 4. Management of Genetic Obesity

There are three therapeutic categories to treat obesity: lifestyle modification, medical treatment, and bariatric surgery. The role of genetic factors in obesity is not only a risk factor but also affects the response to therapeutic options for losing weight based on pharmacogenetics and precision medicine with a multidisciplinary approach. Since hyperphagia is a main clinical feature of monogenic obesity, the most effective management is food restriction. This will need adequate training and involvement of the parents and care providers to prevent early onset obesity. Environmental factors such as physical activity, socioeconomic state, and type of diet could modulate the penetrance of obesity associated with pathogenic mutations to avoid unhealthy environments for these patients [9].

Setmelanotide (Imcivree) is a melanocortin-4 (MC4) receptor agonist used for the treatment of obesity due to proopiomelanocortin (POMC), proprotein convertase subtilisin/keying type 1 (PCSK1), or leptin receptor (LEPR) deficiency. The US Food and Drug Administration approved the drug for chronic weight management in patients 6 years and older with obesity caused by POMC, PCSK1, and LEPR deficiency. Setmelanotide is under consideration for other rare genetic disorders associated with obesity including Bardet–Biedl syndrome, Alstrom syndrome, POMC, and other MC4R pathway heterozygous deficiency obesities. Setmelanotide activates areas in the brain that regulate appetite and fullness, causing patients with specific defects in these areas of the brain not to eat as much and helps to lose weight. It also may increase resting metabolism that can contribute to weight loss. Setmelanotide may lead to weight loss in patients with obesity associated with these conditions but does not treat the genetic defects that cause the condition or other symptoms or signs [93].

The management of adrenal insufficiency is very important with the maintenance of physiologic hydrocortisone replacement in POMC deficiency. Patients with congenital leptin deficiency could be treated by daily injections of recombinant human leptin, which decreases obesity and associated phenotypic abnormalities. Leptin treatment may reduce food intake, fat mass, hyperinsulinemia, and hyperlipidemia in humans, and restores normal pubertal development, endocrine, and immune function [94].

Growth hormone treatment is beneficial in the management of PWS. One of the first comprehensive studies to measure the benefits of growth and body composition with the use of growth hormone on individuals with Prader–Willi syndrome was completed in 1997 by Lindgren et al. This study included 27 affected individuals: 15 with growth hormone treatment for 2 years and 12 with growth hormone treatment for 1 year [95]. They reported that all 27 enrolled individuals showed an increase in height velocity and muscle mass and a decrease in body fat percentage, regardless of time on growth hormone. This study also suggested measurable benefits with growth hormone treatment in regard to a decline in adverse behavioral and psychiatric issues that are associated with PWS [95].

A follow-up study to the aforementioned study was completed by Lindgren et al. in 1998. The intent of this study was to measure and compare the growth and body composition in affected individuals with Prader–Willi syndrome treated with growth hormone in comparison to those not treated. Lindgren et al. found in this study that those treated with growth hormone had an increase in height and a decrease in fat mass and BMI in comparison to those not treated [96].

Another comprehensive study to measure the benefits of growth hormone treatment in affected individuals was completed in 1998 by Eiholzer et al. Twelve affected individuals with Prader–Willi syndrome were enrolled in this study and were grouped and compared based on three different groups: (1) overweight and pre-pubertal, (2) underweight and pre-pubertal, and (3) pubertal. After 12 months of growth hormone treatment within all groups, this study showed a marked increase in growth including height, foot and hand length, and arm span and an increase in lean body mass, muscle mass, and physical performance with increased energy expenditure. They also showed a marked decrease in weight for height, BMI, skin fold thickness, and body fat. Finally, individuals in this study reported to be more active and had increased energy [97].

Whitman et al. also documented similar changes in behavior and physical characteristics with the use of growth hormone treatment in PWS individuals. They noted that the benefits of growth hormone treatment in these patients included having more energy and being more physically fit and demonstrated improvement in memory, sleeping patterns, and social skills [98].

Goldstone et al. in 2008 determined that the highest level of benefit with the treatment of growth hormone to all patients with PWS is similar to those with isolated growth hormone deficiency, including improvement in growth, body composition, and behavior [99]. Festen et al. in 2008 also noted improvement in body composition as one of the most appreciable benefits of growth hormone treatment in affected individuals [100]. Because studies show significant benefits with treatment of growth hormone in individuals with Prader–Willi syndrome, the Food and Drug Administration in 2000 approved injectable somatropin (growth hormone) as a treatment and thus the standard of care for PWS [101]. Similar positive impacts of GH treatment in previously untreated adults with PWS on weight, fat mass, and physical activity levels were also noted by Butler et al. in 2013 [102].

## Figures and Tables

**Figure 1 ijms-23-11005-f001:**
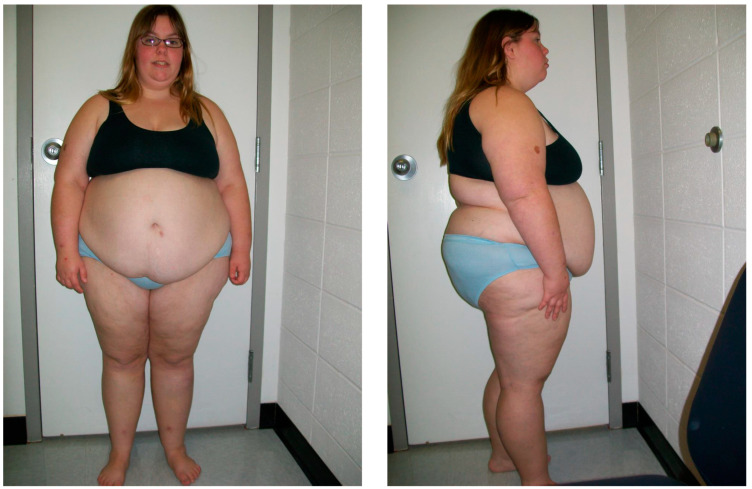
Frontal and profile views of a 16-year-old female with Prader–Willi syndrome due to maternal disomy 15, showing the classical features observed in this obesity-related syndrome.

**Table 1 ijms-23-11005-t001:** Other obesity-related disorders with reported clinical and genetic findings.

Syndrome	Gene	Mode of Inheritance	Clinical Features	Reference
**Borjeson–Forssman–Lehmann syndrome**	*PHF6*	X-linked	Developmental delay Obesity Seizure Skeletal anomalies Large ears Hypogonadism Gynecomastia Distinctive facial features	[37]
**Carpenter syndrome**	*RAB23*	Autosomal recessive	Peculiar facies Brachydactyly of the hands Syndactyly Preaxial polydactyly Congenital heart defects Intellectual disability Hypogenitalism Obesity	[38]
**Cornelia de Lange syndrome**	*NIPBL-CdLS,* *RAD21-CdLS, SMC3-CdLS, BRD4-CdLS,* *HDAC8-CdLS,* *SMC1A-CdLS*	Autosomal dominant X-linked	Microcephaly Synophrys Short nasal bridge Long and/or smooth philtrum Highly arched palate with or without cleft palate Behavioral problems Micrognathia Hearing loss Tendency to overweight	[39]
**CHOPS syndrome**	*AFF4*	Autosomal dominant	Cognitive impairment Coarse facies Heart defects Obesity Short stature, and Skeletal dysplasia.	[40]
**Chudley-Lowry syndrome**	*ATRX*	X-linked	Intellectual disability Short stature Macrosomia Obesity Hypogonadism Distinctive facial features	[41]
**Coffin–Lowry syndrome**	*RPS6KA3*	X-linked	Severe intellectual disability Kyphoscoliosis, Behavioral problems, Progressive spasticity, Paraplegia, Sleep apnea Stroke	[42]
**Kleefstra syndrome**	*EHMT1*	9q34.3 deletion Autosomal dominant	Intellectual disability Obesity Hypotonia Congenital heart defects Genitourinary anomalies Seizures Distinctive facial features	[43]
**Rubinstein–Taybi syndrome**	*CREBBP*, *EP300*	Autosomal dominant	Distinctive facial features, Broad thumbs and halluces Short stature Intellectual disability Obesity in childhood or adolescence	[44]
**Temple syndrome**	Aberrations at the 14q32.2 imprinted region	Maternal disomy 14	Feeding difficulties Hypotonia Motor developmental delay Childhood-onset central obesity Mild facial dysmorphism	[45]

## Data Availability

The data supporting reported material can be obtained upon request from the co-authors.

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
