# Peer review of "Genetics of Obesity in Humans: A Clinical Review"

_ijms, 2022, doi:10.3390/ijms231911005_

Round 1
Reviewer 1 Report
The article is a significant contribution to the field and deals with the very important topic of obesity. The article is well organized and comprehensively described as expected.
Author Response
Prof. Dr. Maurizio Battino
Sep 5/2022
Editor-in-Chief
Department of Odontostomatologic and Specialized Clinical Sciences,
Sez-Biochimica, Faculty of Medicine, Università Politecnica delle Marche,
Via Ranieri 65, 60100 Ancona, Italy
Dear Dr.
Maurizio Battino
We thank you for permitting us to make revisions to further strengthen our manuscript for publication. We appreciate the detailed critiques of the two reviewers and have provided a point-by-point response to the concerns in the cover letter and in the revised manuscript.
Reviewer 1
Open Review
(x) I would not like to sign my review report
( ) I would like to sign my review report
English language and style
( ) Extensive editing of English language and style required
( ) Moderate English changes required
( ) English language and style are fine/minor spell check required
(x) I don't feel qualified to judge about the English language and style
Is the work a significant contribution to the field? |
|
Is the work well organized and comprehensively described? |
|
Is the work scientifically sound and not misleading? |
|
Are there appropriate and adequate references to related and previous work? |
|
Is the English used correct and readable? |
Comments and Suggestions for Authors
The article is a significant contribution to the field and deals with the very important topic of obesity. The article is well organized and comprehensively described as expected.
Response: Thank you for the positive feedback.
Yours Sincerely,
Virginia Kimonis, MD
Univ. of California-Irvine Med. Center
Mail: 101 The City Drive South, ZC4482,
Orange CA 92868
Tel: +1 (714) 456-5791; Fax: +1 (714)456-5330, Pager +1 (714) 506-2063
Email: vkimonis@uci.edu

Reviewer 2 Report
Mahmoud et al. performed an insightful clinical review of the genetics of obesity. I have only a few suggestions to consider that would improve the manuscript.
1) At the first sentence of the last paragraph of the introduction section, the authors wrote:
“New advances in genetic testing have led to identification of obesity-related genes.”
The better term here is perhaps genetic analysis. Genetic testing is usually reserved for genetic counseling, not the performance of a genetic analysis.
2) At the second to the last sentence of the introduction the authors wrote:
“Hence, genetic factors can be divided into three categories: Mendelian (monogenic) syndromic obesity, Mendelian non-syndromic obesity, and polygenic obesity.”
The authors then went on to only discuss the first two of these genetic factors, with no mention of polygenic obesity. If the authors wish to retain their focus on a clinical perspective, and yet treat polygenic obesity at some length, they might perhaps discuss reports of associations of variants in the FTO and MC4R genes with BMI in polycystic ovary syndrome (PCOS) patients (Ewens et al., 2011) and the report by Tu et al. (2017) of an association of a variant in the LEPR gene and PCOS susceptibility. Given that PCOS is a complex polygenic disease (Dumesic et al., 2015; Glueck and Goldenberg, 2019), these associations nicely demonstrate the role of obesity genes in a polygenic context.
3) At the last sentence of the introduction the authors wrote:
“NGS is now in use and emerging as a useful tool to search for candidate genes for obesity in the clinical settings.”
This just seems “tacked on” for lack of a better phrase. Perhaps if it was restated, it may work. A possible rephrasing could be as follows:
While NGS is now in use and emerging as a useful tool to search for candidate genes for obesity in clinical settings, the results of these recent investigations need to be replicated to warrant further consideration, and, as such, results from these studies will not be included in this review.
4) Table 1 just appears with no reference to it in the text. Moreover, it appears to be in the wrong section. It seems like it should be included in Section 3. However, there needs to be a reference to it in the text to warrant inclusion.
References
Ewens KG, Jones MR, Ankener W, Stewart DR, Urbanek M, Dunaif A, Legro RS, Chua A, Azziz R, Spielman RS, Goodarzi MO. FTO and MC4R gene variants are associated with obesity in polycystic ovary syndrome. PLoS One. 2011 Jan 20;6(1):e16390.
Dumesic DA, Oberfield SE, Stener-Victorin E, Marshall JC, Laven JS, Legro RS. Scientific statement on the diagnostic criteria, epidemiology, pathophysiology, and molecular genetics of polycystic ovary syndrome. Endocrine reviews. 2015 Oct 1;36(5):487-525.
Glueck CJ, Goldenberg N. Characteristics of obesity in polycystic ovary syndrome: etiology, treatment, and genetics. Metabolism. 2019 Mar 1;92:108-20.
Tu X, Yu C, Gao M, Zhang Y, Zhang Z, He Y, Yao L, Du J, Sun Y, Sun Z. LEPR gene polymorphism and plasma soluble leptin receptor levels are associated with polycystic ovary syndrome in Han Chinese women. Personalized Medicine. 2017 Jul;14(4):299-307.
Author Response
UNIVERSITY OF CALIFORNIA, IRVINE
UNIVERSITY OF CALIFORNIA, IRVINE MEDICAL CENTER SUITE 800, CITY TOWER
DEPARTMENT OF PEDIATRICS
DIVISION OF GENETICS & METABOLISM 101 THE CITY DRIVE, ZOT 4482
ORANGE, CA 92868-3298
Prof. Dr. Maurizio Battino
Sep 5/2022
Editor-in-Chief
Department of Odontostomatologic and Specialized Clinical Sciences,
Sez-Biochimica, Faculty of Medicine, Università Politecnica delle Marche,
Via Ranieri 65, 60100 Ancona, Italy
Dear Dr.
Maurizio Battino
We thank you for permitting us to make revisions to further strengthen our manuscript for publication. We appreciate the detailed critiques of the two reviewers and have provided a point-by-point response to the concerns in the cover letter and in the revised manuscript.
Reviewer 2
Open Review
(x) I would not like to sign my review report
( ) I would like to sign my review report
English language and style
( ) Extensive editing of English language and style required
(x) Moderate English changes required
( ) English language and style are fine/minor spell check required
( ) I don't feel qualified to judge about the English language and style
Is the work a significant contribution to the field? |
|
Is the work well organized and comprehensively described? |
|
Is the work scientifically sound and not misleading? |
|
Are there appropriate and adequate references to related and previous work? |
|
Is the English used correct and readable? |
Comments and Suggestions for Authors
Mahmoud et al. performed an insightful clinical review of the genetics of obesity. I have only a few suggestions to consider that would improve the manuscript.
1) At the first sentence of the last paragraph of the introduction section, the authors wrote:
“New advances in genetic testing have led to identification of obesity-related genes.”
The better term here is perhaps genetic analysis. Genetic testing is usually reserved for genetic counseling, not the performance of a genetic analysis.
Response: We changed genetic testing to genetic evaluation and analysis in the manuscript as suggested.
2) At the second to the last sentence of the introduction the authors wrote:
“Hence, genetic factors can be divided into three categories: Mendelian (monogenic) syndromic obesity, Mendelian non-syndromic obesity, and polygenic obesity.”
The authors then went on to only discuss the first two of these genetic factors, with no mention of polygenic obesity. If the authors wish to retain their focus on a clinical perspective, and yet treat polygenic obesity at some length, they might perhaps discuss reports of associations of variants in the FTO and MC4R genes with BMI in polycystic ovary syndrome (PCOS) patients (Ewens et al., 2011) and the report by Tu et al. (2017) of an association of a variant in the LEPR gene and PCOS susceptibility. Given that PCOS is a complex polygenic disease (Dumesic et al., 2015; Glueck and Goldenberg, 2019), these associations nicely demonstrate the role of obesity genes in a polygenic context.
Response: We added a paragraph about polygenic obesity to the manuscript.
3) At the last sentence of the introduction the authors wrote:
“NGS is now in use and emerging as a useful tool to search for candidate genes for obesity in the clinical settings.”
This just seems “tacked on” for lack of a better phrase. Perhaps if it was restated, it may work. A possible rephrasing could be as follows:
While NGS is now in use and emerging as a useful tool to search for candidate genes for obesity in clinical settings, the results of these recent investigations need to be replicated to warrant further consideration, and, as such, results from these studies will not be included in this review.
Response: We rephrased the paragraph as suggested by the reviewer
4) Table 1 just appears with no reference to it in the text. Moreover, it appears to be in the wrong section. It seems like it should be included in Section 3. However, there needs to be a reference to it in the text to warrant inclusion.
Response: We moved the table to the correct section, and we refer to it in the text.
References
Ewens KG, Jones MR, Ankener W, Stewart DR, Urbanek M, Dunaif A, Legro RS, Chua A, Azziz R, Spielman RS, Goodarzi MO. FTO and MC4R gene variants are associated with obesity in polycystic ovary syndrome. PLoS One. 2011 Jan 20;6(1):e16390.
Dumesic DA, Oberfield SE, Stener-Victorin E, Marshall JC, Laven JS, Legro RS. Scientific statement on the diagnostic criteria, epidemiology, pathophysiology, and molecular genetics of polycystic ovary syndrome. Endocrine reviews. 2015 Oct 1;36(5):487-525.
Glueck CJ, Goldenberg N. Characteristics of obesity in polycystic ovary syndrome: etiology, treatment, and genetics. Metabolism. 2019 Mar 1;92:108-20.
Tu X, Yu C, Gao M, Zhang Y, Zhang Z, He Y, Yao L, Du J, Sun Y, Sun Z. LEPR gene polymorphism and plasma soluble leptin receptor levels are associated with polycystic ovary syndrome in Han Chinese women. Personalized Medicine. 2017 Jul;14(4):299-307.
Response: We added these references to the reference list.
Yours Sincerely,
Virginia Kimonis, MD
Univ. of California-Irvine Med. Center
Mail: 101 The City Drive South, ZC4482,
Orange CA 92868
Tel: +1 (714) 456-5791; Fax: +1 (714)456-5330, Pager +1 (714) 506-2063
Email: vkimonis@uci.edu
